# The Complex Behaviour of *s*-Process Element Abundances at Young Ages

**Valentina D'Orazi** [1,2,*], **Martina Baratella** [3], **Maria Lugaro** [2,4,5], **Laura Magrini** [6] and **Marco Pignatari** [4,7,8,†]

1 INAF Osservatorio Astronomico di Padova, Vicolo dell'Osservatorio 5, 35122 Padova, Italy
2 School of Physics and Astronomy, Monash University, Melbourne, VIC 3800, Australia; maria.lugaro@csfk.org
3 Leibniz Institute for Astrophysics, An der Sternwarte 16, 14482 Potsdam, Germany; mbaratella@aip.de
4 Konkoly Observatory, Research Centre for Astronomy and Earth Sciences (CSFK), Eötvös Loránd Research Network (ELKH), Konkoly Thege Miklós út 15-17, 1121 Budapest, Hungary; mpignatari@gmail.com
5 Institute of Physics, ELTE Eötvös Loránd University, Pázmány Péter Sétány 1/A, 1117 Budapest, Hungary
6 INAF Osservatorio Astrofisico di Arcetri, Largo E. Fermi 5, 50125 Firenze, Italy; laura.magrini@inaf.it
7 E.A. Milne Centre for Astrophysics, Department of Physics & Mathematics, University of Hull, Hull HU6 7RX, UK
8 Joint Institute for Nuclear Astrophysics, Center for the Evolution of the Elements, Michigan State University, 640 South Shaw Lane, East Lansing, MI 48824, USA
* Correspondence: valentina.dorazi@inaf.it
† NuGrid Collaboration, http://nugridstars.org.

**Abstract:** Open clusters appear as simple objects in many respects, with a high degree of homogeneity in their (initial) chemical composition, and the typical solar-scaled abundance pattern that they exhibit for the majority of the chemical species. The striking singularity is represented by heavy elements produced from the slow process of the neutron-capture reactions. In particular, young open clusters (ages less than a few hundred Myr) give rise to the so-called barium puzzle: that is an extreme enhancement in their [Be/Fe] ratios, up to a factor of four of the solar value, which is not followed by other nearby s-process elements (e.g., lanthanum and cerium). The definite explanation for such a peculiar trend is still wanting, as many different solutions have been envisaged. We review the status of this field and present our new results on young open clusters and the pre-main sequence star RZ Piscium.

**Keywords:** stars: abundances; stars: AGB and post-AGB; stars: pre-main sequence; stars: individual: RZ Psc; Galaxy: abundances; Galaxy: open clusters and associations: general; Galaxy: solar neighbourhood

## 1. Historical Background

Open clusters (OCs) are classically considered nature's closest approximation to simple stellar populations. In general, they consist of stars born at the same time and having the same initial chemical composition, e.g., [1]. OCs have been used to scrutinise the chemical properties of the Galactic disc and its evolution ([2–4]). A large number of spectroscopic stellar data (low and intermediate/high resolution, roughly ranging from R ≈ 10,000 up to ≈100,000) have been collected over the years. Recent observational campaigns involve the Open Cluster Abundances and Mapping (OCCAM) survey, based on the SDSS/APOGEE survey [5], the Gaia-ESO survey [6], Open Clusters Chemical Abundances from Spanish Observatories (OCCASO, [7]), and GALactic Archaeology with Hermes (GALAH, [8]). This unprecedented wealth of spectroscopic data provided us with compelling evidence that: (i) clusters are chemically homogeneous (at the level of 0.02–0.03 dex, [9,10], unless evolutionary effects [11] or planetary engulfment episodes have occurred, e.g., [12,13]). (ii) OCs define the well-known Galactic radial metallicity gradient (e.g., [14] and references therein), and lack an age-metallicity relationship, so that it is more important where the clusters formed rather than when. (iii) OCs exhibit a solar-scaled chemical composition

as far as $\alpha$-, odd/even, and iron-peak elements are concerned. Conversely, they have proven to reveal a very peculiar abundance pattern in terms of heavy elements. Elements heavier than iron are primarily synthesised via subsequent capture of neutrons: these processes may be distinguished in rapid ($r$) or slow ($s$), defined according to the $\beta$-decay time scale. Although the unequivocal astrophysical site for the $r$ process is unknown [15], among different scenarios, binary neutron star mergers (see, e.g., [16]) and magneto-rotationally driven supernovae (e.g., [17]) emerge in the last decade as the most promising astrophysical sources. On the other hand, the *main* and the *strong* component of the $s$ process ([18–22]) happen in thermally-pulsating low-mass ($\approx$1.5–4 $M_\odot$) asymptotic giant branch stars (AGBs). The *weak s*-process component (forming elements up to A$\sim$90) befalls massive stars during convective He core and C shell burning stages ([23,24]). The $s$-process production from massive stars and from AGB stars is responsible for around half of the abundance of the heavy elements beyond Fe [18].

Strontium, yttrium, and zirconium comprise the first peak of the $s$ process in the solar abundance distribution; barium, lanthanum, and cerium belong to the second peak, whereas lead defines the end of the third peak.

In 2009, D'Orazi and colleagues discovered perhaps the most interesting chemical feature characterising Galactic OCs: there is a significant increasing trend of barium abundances as a function of the cluster age [25]. This trend was observed in both dwarf and giant stars (although with a small offset between the two samples). In particular, OCs with ages less than $\sim$150–200 Myr can reach values as large as $\approx$0.5–0.6 dex in their [Ba/Fe] (that is more than a factor of four of the solar abundance). In addition, and vice versa, the old(er) cluster counterparts ($t \gtrsim 1$ Gyr) agree with a solar-scaled abundance pattern. In that work, we suggested a supplementary efficiency of the $s$-process element production by low-mass AGBs: in order to reconcile theory and observations, the stellar yields for AGBs with masses of $1-1.5$ M$\odot$ have to be increased by a factor $\sim$6. However, this explanation would have been able to reproduce only a modest enhancement in the intermediate-age clusters, but failed to explain the level of 0.6 dex observed in very young clusters (see that paper for more details). Following that investigation, several works substantially confirmed the trend for Ba abundances in both clusters ([26–31]) and field stars ([32]). Another puzzle highlighted by [30] is the raising trend of Ba with age combined with the different behaviour of La, generating an observed [Ba/La] ratio not compatible with an $s$-process signature. Because of this signature, Ref. [30] suggested that the increase in Ba was a Galactic Chemical Evolution (GCE) signature of the intermediate neutron-capture process or $i$-process [33], and not a product of the $s$-process. In general, at variance with the general consensus on the increasing Ba trend, the behaviour of the other $s$-process elements have been matter of debate in the last 10 years. Contradictory results have been presented in the literature [28–30,34–36]. On the theoretical side, different interpretations encompass deviations from local thermodynamical equilibrium (LTE) for Ba II lines, the activation at young age of the so-called $i$-process [30,33], chromospheric activity, and intense magnetic fields [8,37]. Interestingly enough, none of them is able to reproduce and fully take into account the observed pattern for barium and first- and second-peak (e.g., La and Ce) $s$-process elements.

We recently published new results on this topic that will be presented in the following section, by providing the reader with an updated broad context view of the field. In addition, we present a re-analysis of star RZ Piscium, a G4 pre-main sequence star: Shen and collaborators [38] claim a solar Ba abundance ([Ba/Fe] = +0.18 ± 0.15), despite its young age of $\approx$20–30 Myr.

## 2. Recent Updates

Recent updates and new results in the field of $s$-process element abundances in young stellar populations have been recently published by our group (Baratella et al. (2021, [39]). We analysed a sample of five young OCs (IC 2391, IC 2602, IC 4665, NGC 2516, and NGC 2547) and one star-forming region (NGC 2264), and presented abundances for

Cu I, Sr I, Sr II, Y II, Zr II, Ba II, La II, and Ce II. The main result was that while Ba is extremely overabundant in young clusters (reaching up to 0.6–0.7 dex), all the other s-process elements follow a solar-scaled abundance pattern. A marginal exception may be the first-peak element Y, which displays small enhancements in [Y/Fe] at about 0.25 dex on average. We introduced as a working hypothesis the impact of strong chromospheric activity and intense photospheric magnetic fields on the strength of spectral lines. The underlying idea is that strong lines such as the Ba II features, which on average form in the upper atmospheric layers, suffer from magnetic intensification. As a result, the shapes of these strong lines are altered, being systematically stronger when compared to a quiet star with similar atmospheric parameters. This would result in providing atypical large abundances.

In the left-hand panels of Figure 1, we show the comparison between the spectra of the sun (light blue line, with rotational profile artificially broadened to match the $v \sin i = 11$ km s$^{-1}$ of the young star) and star 10442256-6415301, a bona-fide member of cluster IC 2602 (∼35 Myr, black dashed line). The spectral regions around Cu I at 5105 Å, Y II at 4883 Å, and Ba II at 5853 Å lines are displayed in the top, middle, and lower panels, respectively.

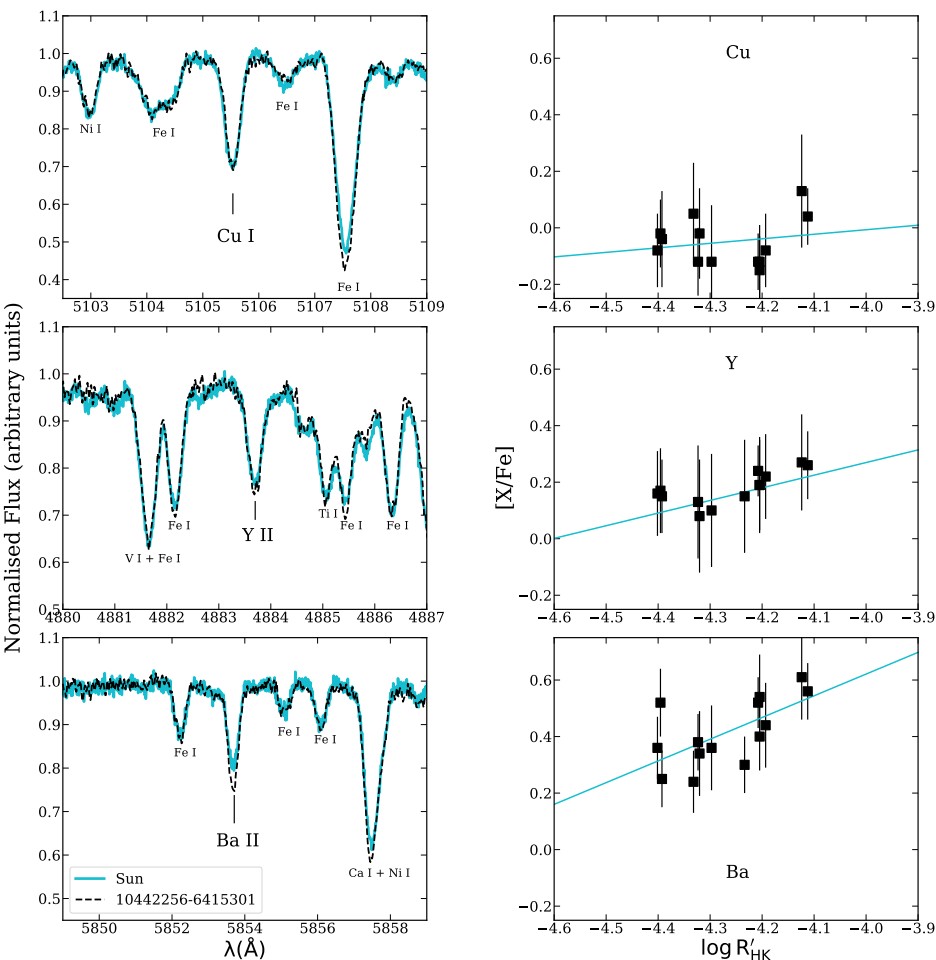

**Figure 1. Left-hand panels**: Comparison of spectra for the sun (light blue line, with rotational profile artificially broadened) and star 10442256-6415301 in IC 2602 (black dashed line, age ∼35 Myr). **Right-hand panels**: [X/Fe] of all stars as a function of their activity index $\log R'_{HK}$, computed as in [40], with a typical systematic error of 0.1 dex. The light blue lines are the linear fits of the measurements.

The comparison clearly demonstrates that Ba II and to a less extent Y II lines are deeper in the young star than in the sun, and this corroborates our derived abundances ([Ba/Fe] = +0.36 ± 0.11, and [Y/Fe] = +0.16 ± 0.15). Conversely, the Cu lines are almost

identical in the two stars ([Cu/Fe] = −0.08 ± 0.13 in the IC 2602 star). We note similar differences also in lines for other elements, such as strong Fe I, Ca I, and Ni I (indicated in the panels). In the right-hand panels of Figure 1, we report the dependency of the derived [X/Fe] of each target as a function of their activity levels represented by $\log R'_{HK}$ ([41], computed as in [40]). While the [Cu/Fe] ratios exhibit no significant dependency on the $\log R'_{HK}$, the correlations for [Y/Fe] and [Ba/Fe] are meaningful. In fact, we obtained a Pearson correlation coefficient of $r = 0.194$ ($p$ value 0.526), $r = 0.598$ ($p$ value 0.024), and $r = 0.634$ ($p$ value 0.015) for Cu, Y, and Ba, respectively.

We note that the activity indicators are not synchronous to our spectra and consequently to our derived abundances. Nevertheless, we conclude that there is in fact a robust indication of a positive correlation between abundances and activity levels, confirming our starting hypothesis (and independent works by, e.g., [8,37])

Unfortunately, the picture is far more complex than this. According to our computation of the line optical depths (i.e., a proxy for where the lines form in the atmospheric layers), the observed behaviour of Y and Ba lines confirmed our initial suggestion ($\log \tau_Y = -2.6$ and $\log \tau_{Ba} = -3.2$). However, the Cu I line forms at a depth similar to Ba ($\log \tau_Y = -3.4$): we should then expect to observe a similar effect. This is not verified (Figure 1). The difference between the Cu and Ba lines can be related to the ionisation stage of the spectral features under scrutiny; per contra, singly-ionised La and Ce lines result in solar abundances. We also investigated the sensitivity to the presence of magnetic fields by looking at the Landé factors $g_L$ of each line: all of them have small similar values (below about 1.3), which cannot account for the detected behaviour.

We also explored the first ionisation potential (FIP) effect [42], according to which coronal abundances obtained from lines with FIP < 10 eV are enhanced with respect to the photospheric values. Our spectral lines are good candidates to display this effect, but all of them have similar FIP values, not explaining the different derived abundances.

From a spectral point of view, we concluded that all the investigated effects may play a role in the derived abundances. Yet, there is no convincing evidence that any of proposed observational effects, or a combinations of them, provide a solution to the Ba puzzle (as Reddy and Lambert [28,29] labelled the anomalous Ba over-abundances of young stellar systems).

From a nucleosynthesis point of view, the adopted GCE models (obtained considering standard stellar yields of each element, see for discussion [39]) are not able to reproduce the observed [Ba/La] time evolution and the massive production of Ba in the last 100 Myr. As expected, the production of Ba and La (both belonging to the second-peak) in the models has to be the same. The only published scenario that may reproduce the Ba production decoupled from La is represented by the *i*-process [30]. However, it is still a mystery what stellar source could host a recent *i*-process production of Ba. For what we know at the moment, the *i*-process could be indeed activated in different types of stars, including low-mass stars, massive stars, and binary systems such as rapidly accreting white dwarfs ([43–48]).

We conclude that the discovery of potential Ba observational issues due to our inability to reproduce well the photosphere of very young active stars could be a promising avenue allowing to solve the Ba puzzle in OCs. At the moment, we cannot fully account for the high Ba enhancement observed in some of the OCs compared to La, but more work is needed before ruling out this possibility. On the other hand, the *i*-process activation observed in stars of young OCs and not, e.g., in the sun is a puzzling scenario, where it is unclear what could be the responsible stellar source.

*The Barium Abundance for RZ Piscium*

RZ Piscium (RZ Psc) belongs to the stellar class of UX Ori type variables, which are Herbig Ae/Be intermediate-mass pre-main sequence stars. The photometric variability phenomenon has been deciphered in terms of obscuration of circumstellar dust in an inclined disc vs. unsteady accretion ([49] and references therein). The young nature of RZ

Psc was established in the comprehensive examination of its atmospheric properties by Potravnov et al. (2014, [50]). The star was ascertained to occupy the transitional region between weak-lined and post-T Tauri stage, with evidence for a warm debris disc, as indicated by the infrared excess beyond 5 μm. Interestingly, Grinin and collaborators [51] suggested an age of $t = 25 \pm 5$ Myr, which is significantly older than the typical dissipation timescale for proto-planetary (gas-rich) discs (e.g., [52]). On the other hand, the presence of gas in the circumstellar environment has been demonstrated from the detection of the NaI resonance doublet lines [50]. In this work, Potravnov and collaborators also presented the metallicity and elemental abundance analysis for several $\alpha$- and iron-peak elements, which seem to express, with the exception of Si and Ca, a metal deficiency at a level of 0.3 dex (see their Figure 5). Subsequently, Potravnov et al. (2019, [53]) published a downward revision of the age estimate based on GAIA DR2 astrometric data [54] and proposed a possible membership of Cas-Tau OB association ($t=20^{+3}_{-5}$ Myr). In 2019, Shen et al. [38] presented Ba abundance for RZ Psc, by adopting spectroscopic parameters and metallicity previously obtained by Punzi et al. (2018, [55]). By analysing co-added HIRES@Keck spectra with a signal-to-noise ratio around 200, these authors found $T_{\mathrm{eff}} = 5600 \pm 75$ K, $\log g = 4.35 \pm 0.10$ dex, $V_t = 2.0 \pm 0.1$ km s$^{-1}$, and [Fe/H] = $-0.11 \pm 0.03$ (error on the mean) dex. The resulting Ba abundance is [Ba/Fe] = $+0.18 \pm 0.15$ dex. Shen et al. concluded that, despite its very young age, the star does not exhibit the typical enhancement in the Ba content. Thus, the Ba puzzle cannot be (merely) related to an age effect.

In this manuscript, we re-analyse the very same spectra but exploit our new approach to chemically characterise young stars, which is thoroughly described in Baratella's series of papers ([39,40,56]. Briefly, our methodology makes use of titanium lines (instead of iron) to derive atmospheric parameters: this causes a considerable revision of the exceptionally large microturbulence velocities ($V_t$ up to 2–2.5 km s$^{-1}$) and sub-solar metallicity achieved for young stars (we refer the reader to our previous papers for further details). Photometric temperatures computed using Casagrande et al. (2010, [57]) and Mucciarelli et al. (2021, [58]) calibrations (adopting VJHK, and Gaia $B_pR_p$ magnitudes, respectively) produce values ranging from $T_{\mathrm{eff}} = 5300 \pm 90$ K (from JK colours) to $T_{\mathrm{eff}} = 5492 \pm 73$ K from $B_pR_p$. We used the average $T_{\mathrm{eff}}$ from these estimates as an input value for our spectroscopic analysis (i.e., $T_{\mathrm{eff}} = 5431 \pm 70$ K). Similarly, we adopted the surface gravity from the GAIA DR3 parallax of $\log g_\pi = 4.30 \pm 0.03$ dex and $V_t = 0.91 \pm 0.05$ km s$^{-1}$ from the relationship by Dutra-Ferreira et al. (2016, [59]).

Our new analysis converges in: $T_{\mathrm{eff}} = 5350 \pm 75$ K , $\log g = 4.05 \pm 0.10$ dex; $V_t = 0.85 \pm 0.15$ km s$^{-1}$ and metallicity from Fe I lines of [Fe/H]I = $-0.01 \pm 0.02$ (EW scatter) $\pm 0.06$ dex (errors due to atmospheric parameters). We detected an indication for the over-ionisation effects of FeII lines as previously found in young clusters ([60–62]), being [Fe/H] II = $+0.16 \pm 0.03 \pm 0.08$ dex (the ionisation balance for iron is obviously not satisfied). Previous atmospheric parameter determinations of RZ Psc have been published by Kaminskii et al. (2000, [63]) and Potravnov et al. (2014, [50]. In particular, Kaminskii et al. derived $T_{\mathrm{eff}} = 5450 \pm 150$ K (from the Boltzmann equilibrium), $\log g = 3.41 \pm 0.02$ dex; $V_t = 2.0 \pm 0.5$ km s$^{-1}$. While $T_{\mathrm{eff}}$ is in fair agreement with our estimate, the surface gravity and micorturbulence velocity are considerably discordant. Conversely, by taking into account observational uncertainties, our atmospheric model complies with the parameters by Potravnov et al., who obtained $T_{\mathrm{eff}} = 5350 \pm 150$ K (by fitting Balmer line profiles), $\log g = 4.2 \pm 0.3$ dex, and $V_t = +1$ km s$^{-1}$. However, they found a sub-solar metallicity ([M/H] = $-0.3 \pm 0.05$ dex), which is in sharp contradiction with standard chemical evolution models of our Galaxy (see, e.g., [64] and Figure 11 in Baratella et al., 2020a [40]). Concerning the comparison with Punzi et al. (2018), despite the reasonable agreement in the resulting metallicity, our values of $T_{\mathrm{eff}}$, $\log g$, and microturbulence are significantly different. We emphasise that a value of $V_t = 2.0$ km s$^{-1}$ as given by Punzi and colleagues does not reproduce the observed line profiles, being too large (we show an example for a wavelength region of $\sim$30 Å around 5080 Å in Figure 2). Once we defined the new set of atmospheric parameters using Ti lines, we then looked into the Ba line behaviour. By using the driver

*synth* of the LTE code MOOG (Sneden [65], 2017 version) and MARCS model atmosphere, we synthesised the Ba II line at 5853 Å (see Figure 3). We found [Ba/Fe] = +0.62 ± 0.14 dex. The $V_t$ estimate is responsible for the discrepant value between our study and Shen et al. and substantially resolves the difference in [Fe/H] and [Ba/Fe] for star RZ Psc. Stellar parameters and abundances for Fe, Ti, and Ba are reported in Table 1. Additionally, differences in line-lists and in the adopted procedure to minimise spurious trends partially contribute to the disagreement with our parameter and abundance determination.

To summarise, our findings confirm the anomalous Ba behaviour for young stars (see Figure 4). Consequently, at the current stage Ba cannot be used as a reliable tracer of the *s* process at young age in our Galaxy, unless either depth-dependent $V_t$ values or 3D atmospheric models are employed.

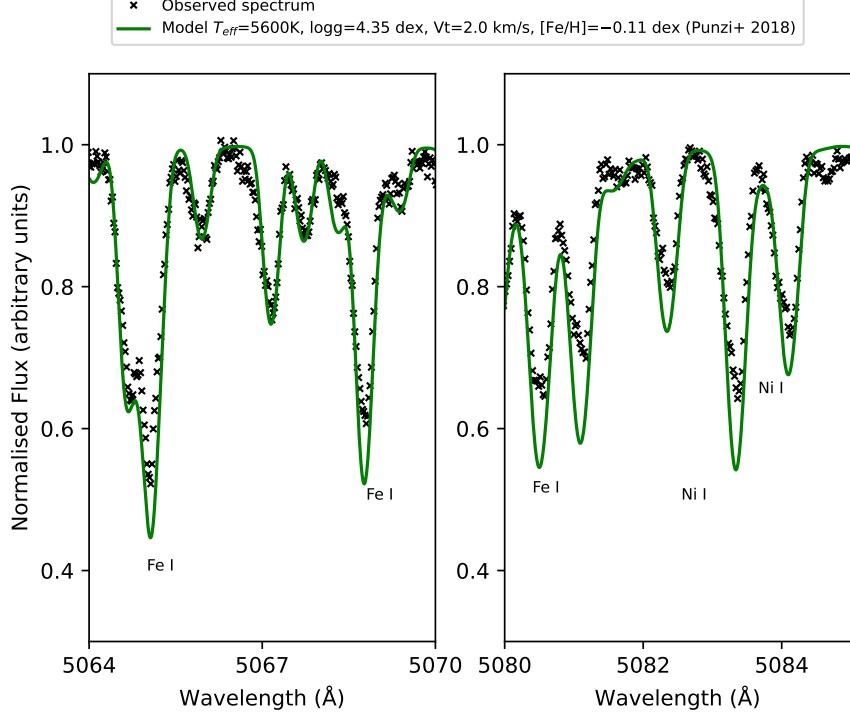

**Figure 2.** Comparison between observed and synthetic profiles calculated by adopting the atmospheric models by Punzi et al., 2018. Several strong iron and nickel lines are labelled. The value of $V_t = 2$ km s$^{-1}$ is too large to successfully reproduce the observed line profiles.

**Table 1.** Results of our analysis for RZ Pic. Errors on abundances include EW/spectral synthesis uncertainties ($\sigma$1) and errors on stellar parameters ($\sigma$2).

| Star | $T_{\text{eff}}$ (K) | log $g$ (dex) | $\xi$ (km/s) | [Fe/H] | [Ti/H] | [Ba/Fe] |
|---|---|---|---|---|---|---|
| RZ Pic | 5350 ± 75 | 4.05 ± 0.10 | 0.85 ± 0.15 | −0.01 ± 0.02 ± 0.06 | −0.03 ± 0.03 ± 0.11 | +0.62 ± 0.14 ± 0.10 |

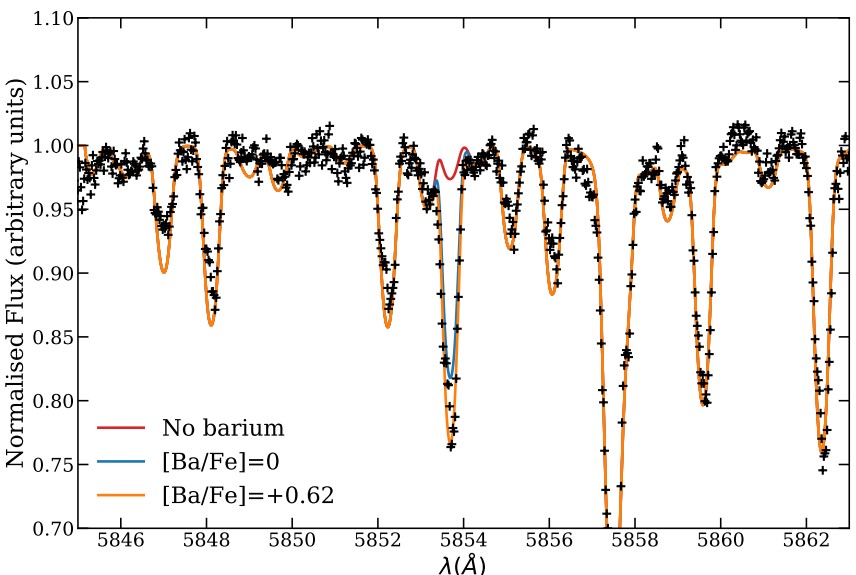

**Figure 3.** Comparison between observed (black crosses) and synthetic profiles around the Ba II line at 5853 Å for star RZ Psc.

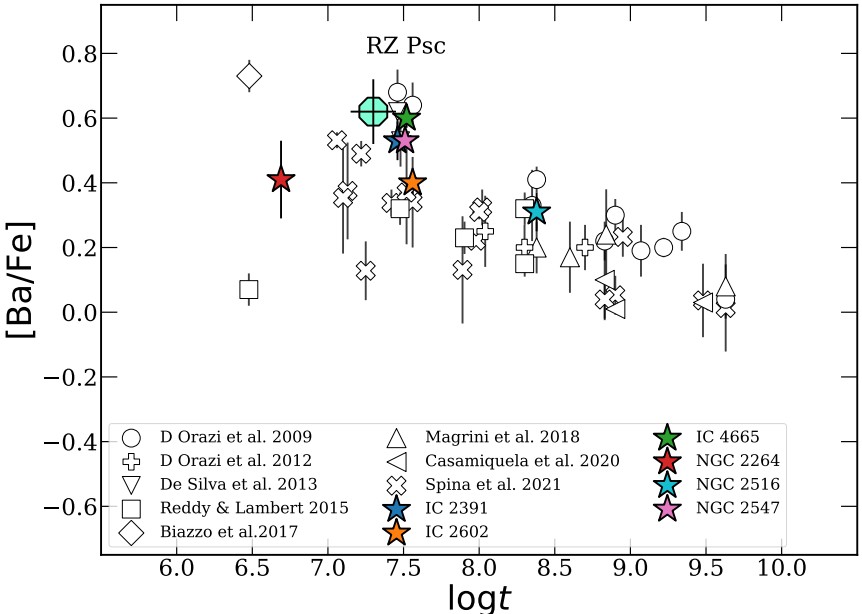

**Figure 4.** [Ba/Fe] ratios as a function of age (time in years) for clusters analysed in Baratella et al., 2021, along with previous results. The new determination of RZ Psc is labelled as a cyan octagonal symbol. For RZ Psc we adopted an age of $20 \pm 5$ Myr, which includes the different range reported in the literature (see discussion in the main text). The ages reported for OCs come from a single source, as detailed in Baratella et al. 2021 [39]. Credit: Figure adapted from Baratella et al. A&A, 653, A67, 2021, and reproduced with permission © ESO.

## 3. Conclusions and Future Perspectives

Recent investigations in the field of heavy-element abundances at young age in our galaxy corroborate the preliminary discovery by D'Orazi et al. (2009). In fact, OCs and associations with ages younger than ≈200 Myr display an extreme overabundance in

their [Ba/Fe] ratios, likely not accompanied by other first and second peak *s*-process elements (with the possible exception for a modest increase in Y abundances). Manifold justifications have been introduced, which include deviations from the LTE approximation and/or the activation of the *i* process. At the moment, a true nucleosynthesis origin of the Ba enhancement due to the *i* process activation in a not defined stellar source is the only explanation for the available observations. However, such an anomalous behaviour compared to what we observe in the sun provides additional puzzles to fit within the context of our understanding of the GCE of the Milky Way disk.

Although the understanding of the physical reason for Ba enhancements is a major goal of our analysis, this specific feature can be exploited as a diagnostic of stellar youth and provide an independent age estimate for field stars. This is particularly useful in the case of planet-host stars, because the adopted age impacts the estimated mass of the sub-stellar companion (the better you know the star, the better you know the planet). This has been proven to be successful in several recent studies from our group focused on, e.g., GJ 758 and the curious case of GJ 504; the latter a very likely example of a rejuvenated star (i.e., old but active, see discussion in [66]).

To conclude, we promote the use of lanthanum as a reliable tracer of the s-process element content of the present-day solar neighbourhood; forthcoming instrumentation (very high resolution multi-object spectrograph including the blue regions of the wavelength domain) will furnish fundamental information in this respect. Finally, we caution that abundance ratios such as, e.g., [Y/Mg] and/or [Y/Al] (see, e.g., [67] ) should be carefully used as "chemical clocks" for systems younger than the Hyades, since Y abundances can be artificially (slightly) enhanced at young ages.

**Author Contributions:** Conceptualization, V.D., M.B., M.L., L.M. and M.P.; methodology, V.D., M.B.; software, M.B.; validation, V.D., M.B.; formal analysis, M.B.; investigation, V.D., M.B., M.L., L.M. and M.P.; resources, V.D.; data curation, V.D., M.B., L.M.; writing—original draft preparation, V.D., M.B., M.L., L.M. and M.P. All authors have read and agreed to the published version of the manuscript.

**Funding:** This research was partially funded by PRIN-INAF 2019 "Planetary systems at young ages (PLATEA)", and PRIN-INAF 2019 "Feasibility Study for an optical IFU on Extreme-AO instruments: ELVIS + SHARK-VIS at LBT".

**Data Availability Statement:** Data presented in this work are publicly available through the ESO archive (https://www.archive.eso.org, (accessed on 1 October 2021)) and Keck archive (https://www2.keck.hawaii.edu/koa/public/koa.php, (accessed on 1 October 2021)).

**Acknowledgments:** This research has made use of the Keck Observatory Archive (KOA), which is operated by the W. M. Keck Observatory and the NASA Exoplanet Science Institute (NExScI), under contract with the National Aeronautics and Space Administration. MP acknowledges support to NuGrid from STFC (through the University of Hull's Consolidated Grant ST/R000840/1), and access to VIPER, the University of Hull HPC Facility. MP also acknowledges the National Science Foundation (NSF, USA) under grant No. PHY-1430152 (JINA Center for the Evolution of the Elements). MP and ML thank the "Lendulet-2014" Program of the Hungarian Academy of Sciences (Hungary), the ERC Consolidator Grant funding scheme (Project RADIOSTAR, G.A. n. 724560, Hungary), the ChETEC COST Action (CA16117), supported by the European Cooperation in Science and Technology, the ChETEC-INFRA project funded from the European Union's Horizon 2020 research and innovation programme (grant agreement No 101008324), and the IReNA network supported by NSF AccelNet. This work made extensive use of NASA ADS, Simbad, Vizier, and arXiv databases.

**Conflicts of Interest:** The authors declare no conflict of interest.

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
