# Peer review of "The Complex Behaviour of s-Process Element Abundances at Young Ages"

_universe, doi:10.3390/universe8020110_

Round 1

Reviewer 1 Report

I enjoyed reading this paper and recommend its publication with a few suggestions that I hope the authors are amicable to including in their final draft.

I have broken these comments into three categories:  minor edits, suggestions that I hope to see the authors implement in the final publication, collegial suggestions the authors can take or dismiss at their discretion.

Minor edits:

  1. Line 115 singular/plural “However, the Cu I line form at depth…”

  2. Line 226.  Space in “iprocess”

Issues that I recommend the authors address in their final draft:

  1. Section 1, paragraph 1:  It would be useful to inform the reader of the types of stars that are used to measure the abundances for the open clusters (i.e. solar type stars).

  2.  

    Please provide a citation for the log R’_hk statistic.  Linsky et al. 1979?

  3. Figure 3.  What is the green line in Figure 3?  Is green the color that results from the combination of the other three colors?  Is it the same as the green line in Figure 2?  It would be useful to say which Ab(Ba) values in the legend for the plot correlate with which [Ba/Fe] values discussed in the text.  Consider making this plot more friendly to printing in black & white or greyscale.

  4. Figure 4.  Make it clear in the caption what “t” is in “log t” in the x-axis label.  Is it time in years?

Collegial suggestions the authors can take or dismiss at their discretion:

  1. Section 2:  A few readers may wonder why the destruction/depletion of Barium in the atmospheres of stars as they age (like happens to in some stars with Lithium) is not considered to explain the Barium enhancement in younger clusters.  This may be obvious to experts, but other readers may recall this phenomenon with Lithium and wonder why it is not considered for Barium.

  2. Section 2:  This is another less likely possibility but has anyone done work to rule out/eliminate cosmic ray spallation (x-process) as a possible contributor to this enhancement?

  3. Section 2.2:  If space allows a table of atmospheric parameters and abundances derived using them for RZ Psc may be useful to help the reader digest that information.

    I found section 2.2 very informative as a demonstration of how sensitive the Barium abundances can be to atmospheric model parameters.

  4. It might be useful to introduce Figure 4 earlier in the paper as summarizing the work described in Section 1.

Author Response

FIle attached.

Reviewer 2 Report

This paper is a contribution to a special issue to the journal "Universe", with the overall theme being in honor of Maurizio Busso's 70th birthday.  The paper is good and worthy of publication in this journal, and one does not expect an extended manuscript here.  Nonetheless, the paper has an intriquing small new result that will be of interest to many others.

THe paper is generally well-written and I have only minor comments listed below.  I look forward to seeing the paper published.

COMMENTS

The reference list is awkward and sometimes missing journal names

lines 28:29

"... lack an age-metallicity ..."

    I'M NOT SURE THAT THIS IS EXACTLY WHAT NETOPIL+2022 SAY; PLEASE RECHECK.

line 46: 

"... defines the third peak."

    DEFINES THE END OF THE THIRD PEAK? (PB AND BI ARE THE LAST STABLE ELEMENTS,

RIGHT?

lines 85-86:

"The marginal exception is played by [Y/Fe] ratios, which display small but

significant enhancements at about ≈0.25 dex level." 

    A LITTLE AWKWARD PHRASING; HOW ABOUT

"A marginal exception may be the first-peak element Y, which displays

small enhancements in [Y/Fe] at about 0.25 dex on average."

    OR SOMETHING SIMILAR

lines 99-100:

"... clearly demonstrates that Y II and Ba II lines are deeper ..."

    THE EFFECT DISPLAYED IN THE FIGURE IS MARGINAL FOR Y; PLEASE RE-PHRASE TO

SOFTEN THIS STATEMENT

lines 130-131:

"... the adopted Galactic Chemical Evolution (GCE) models ..."

    A REFERENCE WOULD BE GOOD HERE

lines 136-137:

"... what stellar source could host such a recent i-process production

peaked at Ba, which is not yet visible in the solar system."

    THIS IS CONFUSING; PLEASE RE-PHRASE

line 160:

"The authors ..."

    WHICH AUTHORS?  CLARIFY FOR THE READER

line 148 and elsewhere:

"Rz" ---> "RZ"

lines 211-213"

"Additionally, differences in line-lists and in the adopted procedure tominimise spurious trends partially contribute to the disagreement with our

parameter 213 and abundance determination."

    ARE THESE DISCUSSED IN DETAIL ELSEWHERE?  THEY SHOULD BE, GIVEN YOUR VERY

LARGE DIFFERENCE IN BA ABUNDANCE FROM PREVIOUS INVESTIGATIONS

Figure 4:

"... RZ Psc is labelled as a starred symbol ..."

    I THINK YOU MEAN

"... RZ Psc is labelled as a black starred symbol ..."

Author Response

file attached.

Reviewer 3 Report

The paper discusses the elemental abundances in young clusters, which excludes certain origins of chemical enrichment since birth. It focusses on the s-process element, highlighting a puzzle regarding barium. It provides a helpful overview and introduction to what is a complex problem, including discussion of various processes that affect the way abundances are derived from observed line strengths. All of this benefits both scholars studying this topic and those who want to understand it as a context for their own research. The authors also provide new analysis that feeds into the debate. Furthermore, the paper is well written. As such, I recommend it for publication once the comments below have been addressed.

Main points:

Section 1: please clarify whether (or when) you are talking about abundances in main-sequence stars or in the evolved giants. Obviously, chemical peculiarities in main-sequence stars require a very different explanation from those in evolved stars experiencing mixing of their own nucleosynthesis products.

L60: clarify whether the barium enhancement is seen only in young clusters, or there is a trend with age.

In the explanation of the right-hand panels of figure 1 you claim that the copper trend is not, but the yttrium and barium trends are significant. To make such a claim, you will need to quantify the significance. Judging by eye, knowing full well that one or two data points can easily trick the mind, I am not convinced that any of the trends are significant (it doesn’t mean there is no trend, just that it needs more data to be confirmed – or not). This is exacerbated by the fact that you do not indicate the uncertainty on the activity index, so it is unclear how much this “card deck” may be “shuffled” into the order we witness it here.

Around line 128 you suggest not a single of the investigated effects can explain the differences, but what about taken them all together? Might they then explain it?

Figure 2: I cannot judge whether the microturbulent velocity is appropriate – please zoom in on a few line profiles to better see the shape.

Figure 4: RZ Psc is consistent with the other data but near the higher end; Shen et al.’s result (+0.18) would have placed it near the lower end – but still consistent with the data also! You must argue more convincingly why your result for RZ Psc validates the Baratella et al. work but Shen et al.’s determination does not.

Figure 4: you need to say something about the uncertainties in the ages of these objects (clusters, RZ Psc…).

I am not convinced you have shown proof of your statements in the last paragraph of the paper.

Minor corrections:

L16: “consists” -> “consist”

L19: remove the hyphens at the end of “low” and “high” or join the latter with “resolution”

L19: “interemediate” -> “intermediate”

L20: the comma at the start should be moved to the end of the preceding line

L25: “data-set” -> “data”

L26: “effect” -> “effects”

L40/41: “befalls in massive” -> “befalls massive”

L51: “counterpart” -> “counterparts”

L59: separate “)and”

L63: the acronym “GCE” is not introduced until line 131.

L68: “recent” -> “young”

L70: “Interesting” -> “Interestingly”

L87: “about ~” -> “about” or “~”

Figure 1 caption: last sentence should end with a full stop.

L115: “line form” -> “line forms”

L121: “take into account” -> “account for” (?)

L132: “to reproduced” -> “to reproduce”

L130-133: provide one or more references for this statement.

L146: “not e.g.,” -> “not, e.g.,”

L147: what it could be” -> “what could be”

L154: “Potravnonv” -> “Potravnov”

L161/162: the dashes used the separate the parts of the sentence should be isolated from the words (as you have done correctly elsewhere).

L165 “to” -> “of”

L167: explain the acronym “SNR”

L172: start a new paragraph when you start talking about your re-analysis.

L174: “series papers” -> “series of papers”

L189: “FeII” -> “Fe II”

L191: “determination” -> “determinations”

L192: the “I” with the accent at the end of “Kaminskii” needs to have its dot removed (in LaTeX: \i ) (two occurrences on this line)

L193: “Boltzman” -> “Boltzmann”

L202: “value” -> “values”

L206: “than” -> “then”

L216: typeset the “t” in “Vt” in subscript.

Figure 2 caption: “caluclated” -> “calculated”

Figure 2 caption: typeset the “Vt” and “s-1” correctly.

Figure 3 caption: “BaII” -> “Ba II”

Figure 3 caption: separate the Angstrom sign from the word that follows.

Figure 4: as there are already starred symbols used for some clusters, it would be clearer to use a different symbol for RZ Psc (or for those clusters).

Figure 4 caption: end the last sentence with a full stop.

L226: “iprocess” -> “i process”

L227: “behavior” -> “behaviour” (at least to be consistent with the spelling used in the title)

L228: “provide” -> “provides”

L231: “a diagnostics” -> “a diagnostic”

L235: “studied” -> “studies”

L236: “on e.g.,” -> “on, e.g.,”

L263: I am puzzled by this statement. How can you have obtained consent from RZ Psc, for instance? Would you need to?

Author Response

file attached
